# A clinical deep learning framework for continually learning from cardiac signals across diseases, time, modalities, and institutions

Dani Kiyasseh [1 ✉], Tingting Zhu[1] & David Clifton[1]

Deep learning algorithms trained on instances that violate the assumption of being independent and identically distributed (i.i.d.) are known to experience destructive interference, a phenomenon characterized by a degradation in performance. Such a violation, however, is ubiquitous in clinical settings where data are streamed temporally from different clinical sites and from a multitude of physiological sensors. To mitigate this interference, we propose a continual learning strategy, entitled CLOPS, that employs a replay buffer. To guide the storage of instances into the buffer, we propose end-to-end trainable parameters, termed task-instance parameters, that quantify the difficulty with which data points are classified by a deep-learning system. We validate the interpretation of these parameters via clinical domain knowledge. To replay instances from the buffer, we exploit uncertainty-based acquisition functions. In three of the four continual learning scenarios, reflecting transitions across diseases, time, data modalities, and healthcare institutions, we show that CLOPS outperforms the state-of-the-art methods, GEM[1] and MIR[2]. We also conduct extensive ablation studies to demonstrate the necessity of the various components of our proposed strategy. Our framework has the potential to pave the way for diagnostic systems that remain robust over time.

[1] Department of Engineering Science, University of Oxford, Oxford, UK. ✉email: dani.kiyasseh@eng.ox.ac.uk

Cardiac arrhythmia diagnosis, the identification of abnormalities in the functioning of the heart, is instrumental in guiding the decision-making process of both cardiologists and clinicians at large. To perform such a diagnosis, it is common to leverage the electrocardiogram (ECG), a signal that measures the electrical activity of the heart. The advent of deep-learning systems allows for automated cardiac arrhythmia diagnosis at scale and with reasonable accuracy. Many of these systems require that data are independent and identically distributed (i.i.d.), and, as such, are developed based on a single snapshot of cardiac data. In other words, such deep-learning systems are static. The violation of the i.i.d. assumption, which can be detrimental to the learning behavior of a system, may arise, for example, when data are streamed sequentially from a sensor or from multiple sensors in a dynamic environment. In clinical settings, this dynamic environment is reflected by the multitude of physiological sensors that generate time-series recordings that may vary temporally (due to seasonal diseases; e.g., the flu), across patients (due to different hospitals or hospital settings), and in their modality. Regardless of the setting, such dynamics result in a shift in the distribution of data, arguably the "Achilles heel" in the deployment of deep-learning systems[3].

Tackling the challenges posed by dynamic environments is the focus of continual learning (CL) whereby a learning system, when exposed to tasks in a sequential manner, is expected to perform well on current tasks without compromising performance on previously seen tasks. The outcome is a single system that can reliably solve a multitude of tasks. The dynamic and chaotic environment that characterizes healthcare necessitates systems that are dynamically reliable; those that can adapt to potential data distribution shift without catastrophically forgetting how to perform tasks from the past. Such dynamic reliability implies that systems are less likely to require re-training on data or tasks to which it has been exposed in the past, thus reducing their overall data requirements and the burden placed on researchers. Furthermore, we hypothesize that clinical deep-learning systems that perform consistently well over time and across a multitude of tasks are more likely to be trustworthy, a desirable trait sought after by medical professionals[4].

Continual, or lifelong, learning algorithms have achieved notable success in the field of computer vision. Such algorithms, a recent summary of which is provided by Parisi et al.[5], belong predominantly to one of three approaches; those based on neural architecture changes, regularization, or replay buffers. The latter approach, shown to be preferable to the remaining two, involves storing data points into, and acquiring them from, a buffer during the learning process. Replay buffers have manifested in various forms[1,6–8]. For example, in gradient episodic memory (GEM)[1], a replay buffer is naively populated with the last $m$ data points observed in previous tasks. Isele et al.[9] and Aljundi et al.[10] employ a more sophisticated strategy where a quadratic programming problem is solved to identify suitable data points for storage. Another method, maximally interfered retrieval (MIR)[2], stores data points into a buffer using reservoir sampling. It also acquires data points from a buffer based on whether a system incurs a large change in the loss when classifying such data points, given subsequently updated parameters. Such an approach is computationally expensive since it requires multiple forward and backward passes through the deep-learning system per batch of instances. As for replaying data points from the buffer, Titsias et al.[11] and Nguyen et al.[12] exploit the notion of model uncertainty and variational methods. Beyond computer vision, CL in the medical domain has been limited to the application of existing methodologies on a chest X-ray dataset[13]. However, to the best of our knowledge, no study has designed and evaluated a continual deep-learning system in the context of physiological signals.

More formally, we designed and evaluated a continual deep-learning system capable of diagnosing cardiac arrhythmias based on ECG data streaming in a sequential manner. The system received an input of single-lead ECG data and returned a single cardiac arrhythmia diagnosis. We hypothesized that such a deep-learning system could perform the clinical task of cardiac arrhythmia diagnosis in several dynamic environments without catastrophically forgetting how to perform previous tasks.

In contrast to previous research that investigates the storage of data points into, and the acquisition of data points from, the buffer independently, we focused on a dual storage and acquisition strategy. First, to determine which data points were most informative for storage, we associated each one with a learnable parameter that acted as a coefficient of the loss function. We showed that this parameter is a proxy for the difficulty with which a data point is classified by the deep-learning system, lending itself to a high degree of intepretability. Second, to determine which data points should be replayed from the buffer, we periodically quantified the uncertainty with which each data point was classified by the deep-learning system. When combining the buffer storage and acquisition mechanisms, we showed that our CL framework outperformed state-of-the-art CL methods, including GEM and MIR, in three of the four diverse CL scenarios. We also conducted extensive ablation studies and showed that our proposed buffer storage and acquisition mechanisms were essential for the improved performance. Furthermore, we validated our interpretation of the learnable parameters as a proxy for difficulty by making reference to ECG domain knowledge.

## Results

**Data**. We designed and evaluated our deep-learning system using four publicly available datasets comprising ECGs alongside cardiac arrhythmia labels. The first dataset, which we refer to as Cardiology[14], includes ECG data collected via a chest patch from 292 patients alongside twelve cardiac arrhythmia labels: AFIB, AVB, BIGEMINY, EAR, IVR, JUNCTIONAL, NOISE, NSR, SVT, TRIGEMINY, VT, and WENCKEBACH. The second dataset, which we refer to as Chapman[15], includes ECG data collected from 10,646 patients alongside four high-level cardiac arrhythmia labels: AFIB, GSVT, sinus bradycardia, and sinus rhythm. The third dataset, which we refer to as PhysioNet 2020[16], includes ECG data collected from 6876 patients alongside nine cardiac arrhythmia labels: AFIB, I-AVB, LBBB, Normal, PAC, PVC, RBBB, STD, and STE. The fourth dataset, we which we refer to as PhysioNet 2017[17], includes 8528 single-lead ECG recordings alongside four labels: normal, AFIB, other, and noisy. Across all datasets, we split patients randomly into training, validation, and test sets, ensuring that there was no patient overlap between sets.

**Outcomes**. The primary outcome was to diagnose cardiac arrhythmias in various CL scenarios, defined as environments with changing dynamics (described next). The secondary outcome was to mitigate catastrophic forgetting experienced by our deep-learning system as a result of data distribution shift. We define catastrophic forgetting as a hindered ability of the deep-learning system to diagnose cardiac arrhythmias based on data exposed to in the past.

**Algorithm development**. The deep-learning system involved a lightweight convolutional network which received a single-lead ECG as input and returned a probability distribution over cardiac arrhythmias as output. This clinical task was performed in four distinct CL scenarios. From hereon forward, we refer to our deep-

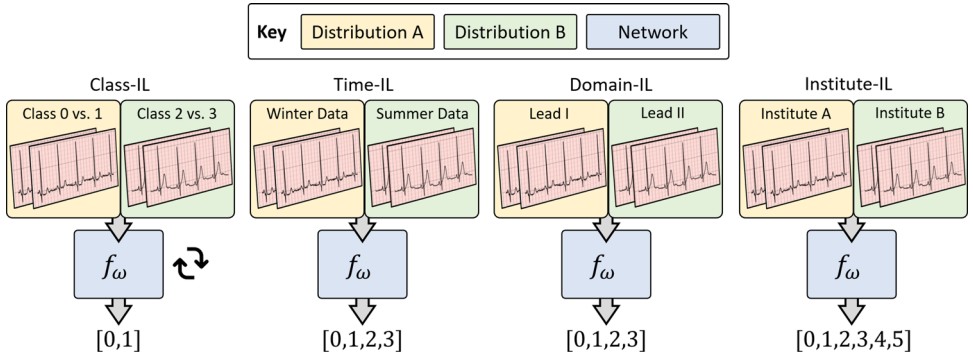

**Fig. 1 Four continual learning scenarios.** A deep-learning system, $f_\omega$, is sequentially exposed to tasks with potential data distribution shifts. In the Class-IL scenario, we present mutually exclusive pairs of classes to reflect novel diseases. In the Time-IL scenario, we present data collected at different times of the year to reflect temporal non-stationarity. In the Domain-IL scenario, we present data from different input modalities to reflect different physiological sensors. In the Institute-IL scenario, we present completely different datasets to reflect disparate healthcare institutions. The system in the Class-IL scenario has a classification head that is specific to the task, and is thus task-specific. In the remaining scenarios, the system is task-agnostic; it is not aware of the task identity of the data.

learning system as CLOPS, for the continual learning of physiological signals.

**Continual learning scenarios**. We simulated four environments with changing dynamics in which the deep-learning system was sequentially tasked with performing cardiac arrhythmia classification.

In the class incremental learning (Class-IL) scenario, the deep-learning system solved a binary classification problem in response to data from mutually exclusive pairs of cardiac arrhythmia classes (see Fig. 1 left). In our context, we split the Cardiology dataset based on the following class-pairs [0, 1], [2, 3], [4, 5], [6, 7], [8, 9], and [10, 11]. In light of this setup, the final classification head of the deep-learning system was specific to each task. This scenario allowed us to evaluate the sensitivity of a network to new classes.

In the time incremental learning (Time-IL) scenario, the deep-learning system solved a multi-class classification problem in response to data collected at different times of the year (e.g., winter and summer) (see Fig. 1 left). In our context, we split the Chapman dataset into three tasks; Term 1, Term 2, and Term 3 corresponding to mutually exclusive dates of the year during which patient data were collected. This scenario allowed us to evaluate the effect of temporal non-stationarity on the system's performance.

In the domain incremental learning (Domain-IL) scenario, the deep-learning system solved a multi-class classification problem in response to inputs with a different modality (see Fig. 1 right). In our context, we split the PhysioNet 2020 dataset according to the 12 leads of an ECG, which can be considered as 12 different projections of the same electrical signal generated by the heart. This scenario allowed us to evaluate the robustness of a system to changes in the input distribution.

In the institute incremental learning (Institute-IL) scenario, the deep-learning system solved a multi-class classification problem in response to inputs from disparate healthcare institutions (see Fig. 1 right). In our context, the notion of disparate healthcare institutions manifested as different datasets. Specifically, the deep-learning system was exposed to the following datasets in sequence: Chapman, PhysioNet 2017, and Cardiology. This scenario allowed us to evaluate the robustness of the system to a change in healthcare institutions, one which impacts both input and output distributions.

**Replay-based method**. In order to satisfy our primary and secondary outcomes, we designed a deep-learning system capable of

diagnosing cardiac arrhythmias accurately while mitigating the phenomenon of catastrophic interference. To achieve this in the aforementioned CL scenarios, we designed a strategy where the deep-learning system identifies important ECG signals, stores them in a buffer, and replays them in the future. By replaying examples from the past, we can reduce the potentially drastic changes in the distribution of the data. In short, our replay-based method is anchored in a buffer-storage and -acquisition mechanism, which are outlined next.

During training, the deep-learning system needs to identify important ECG signals to store in a buffer. To quantify importance, we assigned each ECG signal a parameter (which we refer to as $s$ for storage) that the deep-learning system learns based on data. This parameter represents the difficulty with which an ECG signal is diagnosed according to one of the cardiac arrhythmias. As the deep-learning system concludes its training on a particular task, it stores in the buffer a fraction ($b$) of the ECG signals with the highest importance parameter value. A detailed description and a summary algorithm can be found in the Methods section and Supplementary Note 1, respectively.

When training on tasks in the future, the deep-learning system may be exposed to data from a different distribution. This can hinder the system's ability to perform tasks achieved in the past. To avoid this behavior, the system selectively chooses ECG signals from the buffer and replays them. To achieve this, we assigned a value to each of the ECG signals in the buffer based on the degree to which the system was uncertain about the cardiac arrhythmia diagnosis. By replaying a fraction ($a$) of such uncertain ECG signals from the past, the system was nudged to remember how to diagnose such signals correctly.

**Evaluation**. To evaluate our continual deep-learning system, we exploited metrics common in the field such as average AUC and backward transfer (BWT)[1]. The former evaluates the performance of the deep-learning system at the end of the sequence of tasks it has been exposed to. The latter involves deploying and evaluating the deep-learning system on data exposed to in the past. BWT is a metric that ultimately sheds light on the degree of catastrophic forgetting a system is experiencing.

Our focus on catastrophic forgetting motivated us to propose two additional evaluation metrics. To determine how performance changes "t-steps into the future," we proposed $BWT_t$ that evaluates the performance of the system on a previously seen task, after having trained on $t$ subsequent tasks. We also extended $BWT_t$ to consider all possible time-steps, $t$, and generated the

**Table 1 Performance of various continual learning strategies in the continual learning scenarios.**

| Method | Average AUC (SD) | BWT (SD) | $BWT_t$ (SD) | $BWT_\lambda$ (SD) |
|---|---|---|---|---|
| *Class incremental learning* | | | | |
| MTL | 0.701 (0.014) | – | – | – |
| Fine-tuning | 0.770 (0 .020) | 0.037 (0.037) | −0.076 (0.064) | −0.176 (0.080) |
| Replay-based methods | | | | |
| GEM[1] | 0.544 (0.031) | −0.024 (0.028) | −0.046 (0.017) | −0.175 (0.021) |
| MIR[2] | 0.753 (0.014) | 0.009 (0.018) | 0.001 (0.025) | −0.046 (0.022) |
| CLOPS | **0.796** (0.013) | **0.053** (0.023) | **0.018** (0.010) | **0.008** (0.016) |
| *Time incremental learning* | | | | |
| MTL | 0.971 (0.006) | – | – | – |
| Fine-tuning | 0.824 (0.004) | −0.020 (0.005) | −0.007 (0.003) | 0.010 (0.001) |
| Replay-based methods | | | | |
| GEM[1] | QP problem could not be solved | | | |
| MIR[2] | 0.856 (0.010) | −0.007 (0.006) | −0.003 (0.004) | 0.001 (0.004) |
| CLOPS | 0.834 (0.014) | −0.018 (0.004) | −0.007 (0.003) | 0.007 (0.003) |
| *Domain incremental learning* | | | | |
| MTL | 0.730 (0.016) | – | – | – |
| Fine-tuning | 0.687 (0.007) | −0.041 (0.008) | −0.047 (0.004) | −0.070 (0.007) |
| Replay-based methods | | | | |
| GEM[1] | 0.502 (0.012) | −0.025 (0.008) | 0.004 (0.010) | −0.046 (0.021) |
| MIR[2] | 0.716 (0.011) | −0.022 (0.011) | −0.013 (0.004) | −0.019 (0.006) |
| CLOPS | **0.731** (0.001) | **−0. 011** (0.002) | **−0. 020** (0.004) | **−0.019** (0.009) |
| *Institute incremental learning* | | | | |
| MTL | 0.825 (0.037) | – | – | – |
| Fine-tuning | 0.589 (0.007) | −0.203 (0.022) | −0.097 (0.020) | −0.087 (−0.023) |
| Replay-based methods | | | | |
| GEM[1] | QP problem could not be solved | | | |
| MIR[2] | 0.631 (0.012) | −0.140 (0.015) | −0.018 (0.006) | −0.036 (−0.004) |
| CLOPS | **0.664** (0.015) | **−0.110** (0.016) | −0.063 (0.014) | −0.049 (−0.031) |

For all experiments, the storage and acquisition fractions are $b = 0.25$ and $a = 0.50$, respectively. The mean and standard deviation (SD) are shown across five seeds. MTL refers to multi-task learning and involves training on all tasks simultaneously. GEM could not always solve the quadratic programming (QP) problem. Also note that ↑BWT indicates reduced catastrophic forgetting. CLOPS outperformed GEM and MIR in the Class-IL, Domain-IL, and Institute-IL scenarios (bold).

metric $BWT_\lambda$. This allowed us to identify improvements in the system at the task level. A detailed description of these metrics is provided in the Methods section.

We compared our deep-learning system to several baseline methods. First, we trained a static multi-task learning (MTL) system that had access to all data from the sequential tasks. Second, we trained a naive fine-tuning system that did not deploy an explicit CL strategy. Lastly, we trained two state-of-the-art CL methods, GEM[1] and MIR[2]. A detailed description of these methods is provided in Supplementary Note 3.

In a CL setting, deep-learning systems are evaluated primarily based on generalization performance and the degree to which they experience catastrophic forgetting. We quantified these two components when our deep-learning system was deployed in four distinct CL scenarios, Class-IL, Time-IL, Domain-IL, and Institute-IL, and present these results in Table 1.

When deployed in the Class-IL scenario, our deep-learning system, CLOPS, achieved an AUC = 0.796 (SD 0.013) whereas the MTL paradigm, one that is expected to achieve relatively strong results, only achieved AUC = 0.701 (SD 0.014). One hypothesis for this behavior revolves around the notion of curriculum learning wherein the strategic presentation of sequential tasks to a deep-learning system can contribute to its generalization performance[18] (discussed later in Table 3). Another hypothesis notes that the MTL paradigm in the Class-IL scenario, in contrast to the other strategies, involves a multi-class classification problem. This is arguably harder to solve than a binary classification problem, and, as such, could depress the performance score. On another note, CLOPS not only outperformed state-of-the-art CL methods, GEM, and MIR, in terms of generalization performance but also exhibited constructive interference. For example, CLOPS and MIR achieved an AUC = 0.796 (SD 0.013) and 0.753 (SD 0.014), respectively. They also achieved a BWT = 0.053 (SD 0.023) and 0.009 (SD 0.018), respectively. Such a finding underscores the ability of CLOPS to deal with tasks involving novel classes. We

found that this holds regardless of task order (see Supplementary Fig. 3).

To gain a better understanding of the effect of destructive interference on a deep-learning system, we illustrate, in Fig. 2a, the AUC achieved by a naive, fine-tuning deep-learning system deployed in the Class-IL scenario. As expected, in the absence of an explicit CL strategy, destructive interference was prevalent. For example, the system has quickly forgotten how to perform task [0–1] once exposed to data from task [2–3], both of which constitute binary cardiac arrhythmia diagnosis. Specifically, the system transitioned from an AUC ≈ 0.92 → 0.30 within a matter of a few training epochs. The final performance of the network for that particular task (AUC ≈ 0.78) is also lower than that maximally achieved (AUC ≈ 0.90). In Fig. 2b, we illustrate the AUC of our deep-learning system, CLOPS, showing that it alleviated destructive interference. This can be seen by the absence of significant decreases in AUC and the higher final performance exhibited by the deep-learning system on all cardiac arrhythmia tasks relative to the naive (fine-tuning) strategy.

In addition to changes in medical conditions that may occur within a clinical setting, environmental changes can also introduce seasonal shift into clinical datasets. We quantify the effect of such a shift on deep-learning systems and illustrate, in Fig. 2c, the AUC achieved by a deep-learning system deployed in the Time-IL scenario.

When deployed in the Time-IL scenario, our deep-learning system, CLOPS, performed slightly worse than MIR. For example, CLOPS and MIR achieved AUC = 0.834 (0.014) and 0.856 (0.010), respectively (see Table 1). Despite this outcome, we found that CLOPS continued to exhibit forward transfer (FWT), a feature that indicates that performing a task in the present facilitates the ability of a deep-learning system to perform a task in the future. For example, in Fig. 2d, CLOPS achieved an AUC ≈ 0.62 after a single epoch of training on task Term 3, a value that the fine-tuning deep-learning system, whose results are shown in

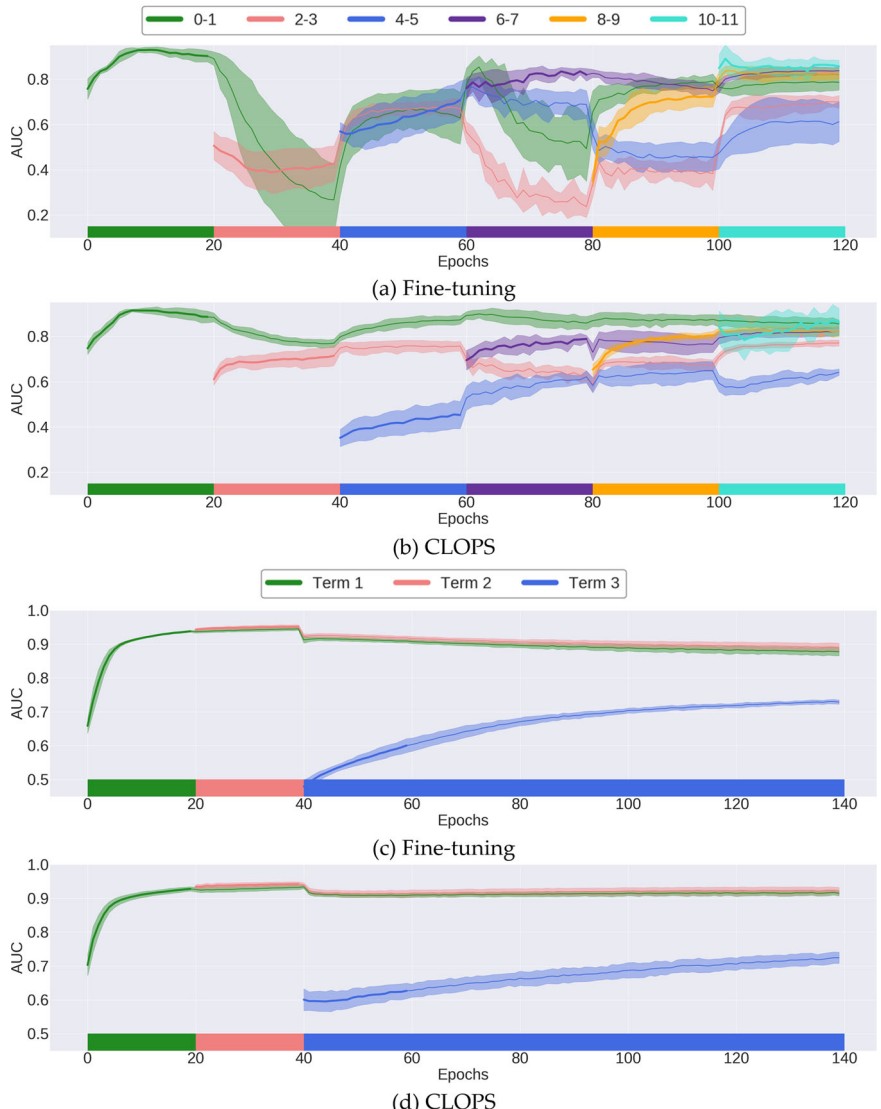

**Fig. 2 Performance of fine-tuning and continual deep-learning system in various continual learning scenarios.** We present the performance of the (**a**) Fine-tuning strategy and (**b**) CLOPS in the Class-IL scenario which is characterized by six sequential tasks. We also present the performance of the (**c**) Fine-tuning strategy and (**d**) CLOPS in the Time-IL scenario which is characterized by three sequential tasks. The x-axis denotes the number of training epochs, the colored blocks reflect the task currently being trained on by the deep-learning system, and the y-axis denotes the average AUC. The performance of the fine-tuning system on tasks not currently being trained on degrades significantly, demonstrating catastrophic forgetting. CLOPS dramatically mitigates this catastrophic forgetting. The results are an average across five seeds and the shaded area represents one standard deviation from the mean.

Fig. 2c, achieved only after 20 full epochs. Such a finding suggests that CLOPS has the potential to reduce the overhead associated with training deep-learning systems. We attribute this FWT to the way in which the deep-learning system identified important ECG signals and modulated the degree to which it should learn from such signals. By placing greater emphasis on more important signals, through a higher-valued task-instance parameter (loss coefficient), the deep-learning system was better able to focus on a subset of ECG signals. An in-depth description of this loss coefficient is provided in the Methods section. It could be argued that such FWT is not unique to CLOPS but rather is a function of the data distribution. In other words, training on data from Term 2 somehow allowed the system to transfer knowledge useful for solving Term 3. Although possible in principle, this was not the case in our experiment. Specifically, if such FWT were attributable to the data distribution, then we would have expected the fine-tuning system to also exhibit some signs of transfer at

epoch 40 in Fig. 2c. However, this was not observed as evident by the starting AUC ≈ 0.50 (random chance) for Term 3. Furthermore, CLOPS mitigated catastrophic forgetting relative to the fine-tuning system. For example, performance on tasks Term 1 and Term 2 was maintained at an AUC > 0.90 when training on task Term 3. This was not observed for the fine-tuning system.

In the Domain-IL scenario, we simulated the presence of various medical sensors by splitting the ECG data into multiple leads, where applicable, and presented those in a sequential manner to the deep-learning system. In Table 1, we present the performance of the deep-learning system when deployed in the Domain-IL scenario. Our deep-learning system, CLOPS, achieved AUC = 0.731 (SD 0.001) whereas a state-of-the-art method, MIR, achieved AUC = 0.716 (SD 0.011). Consistent with earlier findings, CLOPS was also better able to mitigate catastrophic forgetting; CLOPS and MIR achieved BWT = −0.011 (SD 0.002) and −0.022 (SD 0.011), respectively. Furthermore, when deployed in the Institute-IL scenario, our

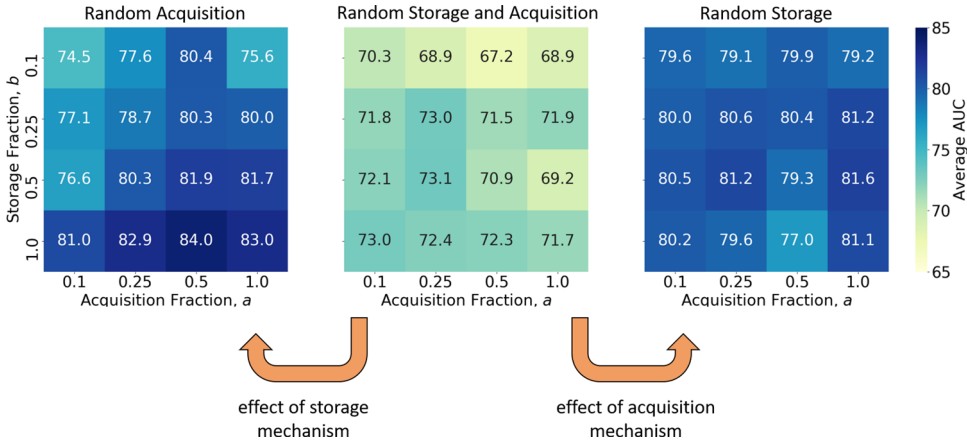

**Fig. 3 Marginal benefit of storage and acquisition mechanisms on performance of continual deep-learning system.** We show three different learning strategies in the Class-IL scenario. Random storage and acquisition stores instances into, and acquires them from, the buffer randomly. Random acquisition stores instances into the buffer using our importance-based strategy and acquires them from the buffer randomly. Random storage stores instances into the buffer randomly and acquires them from the buffer using our uncertainty-based strategy. The results are shown as a function of the storage fraction, $b$, and acquisition fraction, $a$, and are an average across five seeds. Improvement in performance of the random acquisition and random storage learning strategies relative to the random storage and acquisition strategy points to the benefit of our storage and acquisition mechanisms, respectively.

continual deep-learning system outperformed MIR and the fine-tuning system along the dimensions of generalization performance and BWT. For example, CLOPS achieved AUC = 0.664 (SD 0.015) whereas MIR and the fine-tuning system achieved AUC = 0.631 (SD 0.012) and 0.589 (SD 0.007), respectively. Such a finding provides further evidence in support of CLOPS as a favourable replay-based, continual deep-learning system in the context of physiological signals.

**Effect of storage and acquisition mechanisms on performance.** To better understand the root cause of CLOPS' benefits, we conducted additional studies investigating the marginal effect of our storage and acquisition mechanisms on performance. These mechanisms are dependent upon the amount of data that were stored and acquired, and as such, we conducted these studies while varying the fraction of data that are stored into, and retrieved from, the buffer. Such fractions are denoted by $b$ (storage) and $a$ (acquisition), respectively. In the random storage study, we dispensed with our storage mechanism and instead randomly stored ECG signals into the buffer. In the random acquisition study, we dispensed with our acquisition mechanism and instead randomly acquired ECG signals from the buffer. Lastly, in the random storage and acquisition study, we stored ECG signals into, and acquired them from, the buffer randomly. We present the resulting AUC of these experiments in Fig. 3.

Our storage mechanism contributed drastically to the generalization performance of our deep-learning system. Specifically, the incorporation of the storage mechanism increased the AUC of the deep-learning system regardless of the amount of data that were stored and acquired from the buffer. For example, when we only stored 10% of the ECG signals in the buffer ($b = 0.1$) and acquired 50% of the ECG signals from the buffer ($a = 0.5$), we observed an improvement in the AUC = 67.2 → 80.4, reflecting a 13.2% improvement. Such a finding points to how indispensable our storage mechanism is.

When we independently evaluated the acquisition mechanism, we showed that it also contributed drastically to the generalization performance of our deep-learning system. Specifically, the incorporation of the acquisition mechanism increased the AUC of the deep-learning system regardless of the amount of data that were stored and acquired from the buffer. For example, when we only stored 10% of the ECG signals in the buffer ($b = 0.1$) and acquired 10% of the ECG signals from the buffer ($a = 0.1$), we

observed an improvement in the AUC = 70.3 → 79.6, reflecting a 9.3% improvement. Such a finding, particularly with such a small storage and acquisition fraction of ECG signals, points to how robust our acquisition mechanism can be to scarce data environments. Although we presented results illustrating the generalization performance (average AUC), we arrived at similar conclusions when we evaluated the degree to which these mechanisms alleviate catastrophic forgetting (see Supplementary Fig. 10).

**Exploration of alternative storage mechanisms.** Our buffer storage mechanism involved exploiting learnable parameters that acted as a proxy for the difficulty with which an instance was classified by the deep-learning system. In the next two sections, we provide further empirical evidence to support our design of the storage mechanism.

At face value, it may appear that our learnable parameters, given their interpretation as a proxy for instance difficulty, could simply be replaced by the per-instance loss when deciding which instances to store into the buffer. However, we claim that these two quantities, our learnable parameters and the per-instance loss, are not interchangeable. First, this can be seen by the different roles that these two quantities play. Recall that the learnable parameters have a dual role; they act as a coefficient to the loss term (see Eq. (2)), and thus modulate the degree to which the deep-learning system learns from each instance, and they guide the storage of instances into the buffer.

To further substantiate our claim that these two quantities do indeed have different effects on the continual deep-learning system, we conducted the following study. We experimented with a deep-learning system exactly similar to CLOPS, yet comprised a different storage mechanism. Instead of tracking our proposed learnable parameters, we tracked the loss incurred by the deep-learning system on each data point in the current task. When deciding which instances to store in the buffer, we calculated their corresponding area under the loss curve, and identified those with the smallest area, noting that those with a small area coincide with the least difficult data points. In doing so, we remain consistent with the CLOPS implementation that also stored the least difficult instances. In Table 2, we present the performance of such a continual deep-learning system across the CL scenarios.

**Table 2 Effect of the alternative storage mechanism, based on tracking the per-instance loss, on the performance of the continual deep-learning system.**

| CL scenario | Average AUC (SD) | BWT (SD) | BWT$_t$ (SD) | BWT$_\lambda$ (SD) |
|---|---|---|---|---|
| Class-IL | 0.685 (0.006) | −0.004 (0.016) | −0.050 (0.020) | −0.063 (0.026) |
| Time-IL | 0.844 (0.011) | −0.023 (0.013) | −0.011 (0.008) | 0.007 (0.005) |
| Domain-IL | 0.730 (0.007) | −0.021 (0.002) | −0.025 (0.002) | −0.034 (0.004) |
| Institute-IL | 0.615 (0.012) | −0.173 (0.019) | −0.203 (0.014) | −0.278 (0.013) |

The mean and standard deviation (SD) are shown across five seeds. The continual deep-learning system with such a storage mechanism performs worse than that with our proposed strategy (based on task-instance parameters) in the Class-IL scenario. This finding provides evidence in support of our task-instance parameters.

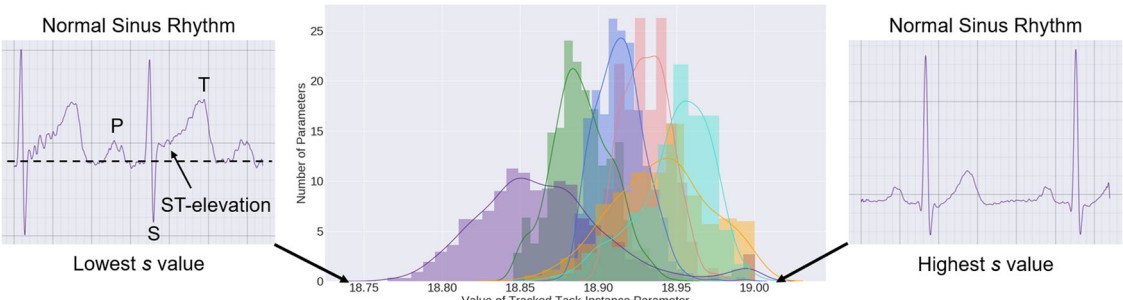

**Fig. 4 Distribution of the storage parameter values, s, learned by the deep-learning system in the Class-IL scenario.** The storage and acquisition fractions are $b = 0.25$ and $a = 0.50$, respectively, and each color corresponds to a different task. Based on our interpretation of storage parameters as a proxy for instance difficulty, we selected two ECG recordings associated with the lowest (most difficult) and highest (least difficult) s values. The ECG recording deemed most difficult by the deep-learning system had a ground-truth cardiologist-derived label of normal sinus rhythm. However, the presence of ST-elevation, a feature typical in an abnormal condition known as myocardial infarction, could have been a source of confusion for the deep-learning system and hindered its ability to correctly diagnose this ECG signal.

By comparing the performance of CLOPS in Table 1 to that in Table 2, we found that the deep-learning system employing task-instance parameters outperformed that employing the per-instance loss in several CL scenarios. For example, in the Class-IL scenario, the two systems achieved AUC = 0.796 (SD 0.013) and AUC = 0.685 (SD 0.006), respectively. This difference is even more notable when looking at BWT. Such an outcome also holds in the Institute-IL scenario, where the two systems achieved AUC = 0.664 (SD 0.015) and AUC = 0.615 (SD 0.012), respectively. On the other hand, in the Time-IL scenario, both systems performed on par with one another when the results are viewed holistically. Although the deep-learning system employing the per-instance loss achieved a slightly higher AUC = 0.844 (SD 0.011) vs. 0.834 (SD 0.014), it performed worse in terms of BWT = −0.023 (SD 0.013) vs. −0.018 (SD 0.004). We arrived at a similar conclusion in the Domain-IL scenario. Overall, these findings indicate that task-instance parameters, despite their simple interpretation as a proxy for instance difficulty, had a more significant positive impact on the continual deep-learning system than the per-instance loss. This justifies our use of task-instance parameters during the learning process. In the next section, we continue to validate these parameters and illustrate additional potential applications.

**Validation of interpretation of storage parameters.** We claimed that our deep-learning system was learning to identify important ECG signals for their eventual storage in a buffer. We then mathematically showed the equivalency of this importance with the difficulty with which the deep-learning system diagnosed the cardiac arrhythmia of ECG signals (Methods section). To validate this claim empirically, we explored and visualized the importance parameters that were learned by the deep-learning system. In

Fig. 4, we illustrate the distribution of these parameters for all ECG signals and across all tasks that the deep-learning system was sequentially exposed to in the Class-IL scenario.

We found that the deep-learning system perceived various tasks to differ in their level of difficulty. For example, the deep-learning system struggled more to solve the cardiac arrhythmia task [6–7] relative to the task [8–9]. This is supported by the observation that the parameter values of the distribution of task [6–7] are lower than those of task [8–9]. At this stage, it might be tempting to correlate the relative difficulty of these tasks to their relative performance, shown earlier in Fig. 2. We believe that such attempts might be of little value given the absence of a strict correlation between loss values and performance scores, such as the AUC. In other words, a decrease in the loss does not always translate to a higher AUC score.

Nonetheless, to validate that these distributions were indeed indicative of the difficulty with which ECG signals were diagnosed, we identified the two ECG signals associated with the lowest and highest importance parameter values and present them alongside the distributions. Based on our setup, these two ECG signals should correspond to the most and least difficult signals to diagnose, respectively. We found that our expectations were indeed corroborated by basic ECG domain expertise. For example, both of these signals had a ground-truth, cardiologist-derived label of normal sinus rhythm. However, the ECG signal deemed most difficult by the deep-learning system exhibited morphological aberrations, such as ST-elevation, a typical feature in certain cardiac abnormalities. Such a feature could have confused the deep-learning system and hindered its ability to diagnose the ECG signal correctly. We provide additional qualitative evidence in Supplementary Figs. 5 and 6. Such a finding reaffirms our interpretation of the importance parameters

as a proxy for the difficulty with which an ECG signal is diagnosed. As a result, we have a tool at our disposal that allows us to monitor the learning dynamics of the deep-learning system and identify data points that the system struggles to diagnose.

In addition to validating our interpretation of the storage parameters qualitatively, we set out to do so more quantitatively. We took inspiration from the curriculum learning literature[18] that has shown that the order with which data are presented to a learning system can impact the system's generalization capabilities. Specifically, we exploited the storage parameters, $s$, to design several curricula based on the notion of task difficulty and similarity, as explained next. First, we fit a Gaussian distribution, $\mathcal{N}(\mu_k, \sigma_k^2)$, to each of the six distributions shown in Fig. 4. Using this information, we defined the difficulty of task, $k$, as $d_k = \frac{1}{\mu_k}$ and the similarity, $S(j, k)$, between task $j$ and $k$ based on the Hellinger distance, as shown in Eq. (1).

$$S(j, k) = 1 - \underbrace{\sqrt{1 - \sqrt{\frac{2\sigma_j\sigma_k}{\sigma_j^2 + \sigma_k^2} e^{-\frac{1}{4}\frac{(\mu_j - \mu_k)^2}{\sigma_j^2 + \sigma_k^2}}}}}_{\mathscr{D}_H = \text{Hellinger Distance}} \quad (1)$$

In Fig. 5, we illustrate the resulting pairwise task similarity matrix for tasks in the Class-IL scenario. For this particular example, we found that task [8–9] is most similar to task [10–11]. Conversely, task [0–1] and [10–11] are identified as being least similar to one another, based on our definition of similarity. Insight from such a similarity matrix, although preliminary, has a twofold effect. First, when coupled with clinical domain expertise, it has the potential to supplement clinical knowledge by potentially identifying differences between medical conditions and patient cohorts, depending on the chosen task definition. Second, by allowing researchers to identify which tasks are believed to be different by the deep-learning system, it could facilitate transfer learning across tasks, domain adaptation, and

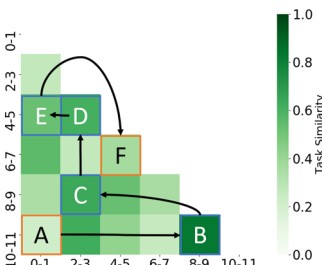

**Fig. 5 Similarity of tasks in the Class-IL scenario overlaid with the chain of tasks used for curriculum learning.** The curriculum begins with the task identified as being the easiest ([10–11]), which is chained to similar tasks by following the arrows, and concludes with the task identified as being the most difficult ([6–7]). The effect of such a curriculum on the learning process can be found in Table 3.

even curriculum learning. To illustrate this point, we experimented with the latter.

We designed a curriculum by first selecting the easiest task ($\downarrow d_k$, task [10–11]), based on Fig. 4, and then creating a chain of tasks that are similar to one another, based on Fig. 5. This chaining process is illustrated by the arrows in Fig. 5. Conversely, for an anti-curriculum, we repeated the process except that we started with the hardest task ($\uparrow d_k$, task [6–7]). In Table 3, we show the performance of the continual deep-learning system as a result of these various curricula.

Our continual deep-learning system achieved the highest constructive interference when trained with a curriculum (easy → hard) as opposed to when trained with an anti-curriculum or randomly ordered tasks. For example, with a curriculum, BWT = 0.087 (0.011) whereas BWT = 0.058 (0.016) and 0.053 (0.023) with an anti-curriculum and randomly ordered tasks, respectively. We hypothesize that transitioning from easy to hard tasks along a chain of similar tasks allowed the continual deep-learning system to efficiently maintain knowledge from one task to the next. Such a finding, which aligns well with the broader expectations of curriculum learning, further supports the intuition that our storage parameters act as a proxy for the difficulty of instances. However, we also found that such improved constructive interference comes at a cost of generalization performance. This was evident by the AUC = 0.744 (0.009) and 0.796 (0.053) achieved by the continual deep-learning system when trained with a curriculum and randomly ordered tasks, respectively. We hypothesize that maintaining knowledge from the past hindered the deep-learning system's ability to perform as well on the current task.

## Discussion

In this paper, we proposed a replay-based CL strategy applied to cardiac signals, entitled CLOPS, with the aim of mitigating destructive interference. CLOPS consisted of an importance-guided buffer storage and an uncertainty-based buffer-acquisition mechanism. In the process, we learned parameters, entitled task-instance parameters, that acted as a proxy for the difficulty with which data points are classified by a deep-learning system. We validated this intuition qualitatively by exploiting ECG domain knowledge and quantitatively by generating learning curricula. Moreover, we showed that, on three of the four CL scenarios, CLOPS outperformed the state-of-the-art methods, GEM and MIR, along both dimensions of generalization performance and backward transfer.

Our system does have several limitations. First our approach assumed that a portion of the data that were used for training could be temporarily stored in a buffer for future use. However, this approach may be considered infeasible due to patient privacy constraints and data storage limitations. Despite these considerations, and in conversations with a medical domain expert, it became apparent that our system could be deployed if data were anonymized and remained within the confines of the same healthcare institution. We appreciate that such insight is

**Table 3 Effect of various curricula on the performance of the continual deep-learning system in the Class-IL scenario.**

| Task order | Average AUC (SD) | BWT (SD) | BWT$_t$ (SD) | BWT$_\lambda$ (SD) |
|---|---|---|---|---|
| Random | **0.796** (0.013) | 0.053 (0.023) | 0.018 (0.010) | 0.008 (0.016) |
| Curriculum | 0.744 (0.009) | **0.087** (0.011) | **0.038** (0.021) | **0.076** (0.037) |
| Anti-curriculum | 0.783 (0.022) | 0.058 (0.016) | −0.013 (0.013) | −0.003 (0.014) |

The storage and acquisition fractions are $b = 0.25$ and $a = 0.50$, respectively. The mean and standard deviation (SD) are shown across five seeds. Bold results reflect the curricula leading to the best performance. The constructive interference exhibited by a deep-learning system following a curriculum based on our task-instance parameters supports our interpretation of such parameters as a proxy for instance difficulty.

institution and country-specific, and as such, recommend that practitioners communicate with their local domain experts before deploying such a system. Moving forward, if our system were to be deployed with other data modalities (e.g., medical images) that more blatantly violate patient privacy, then one could store representations of data points into the buffer, as opposed to the raw data points themselves. Another option would involve incorporating concepts of differential privacy[19] into the continual deep-learning system. Second, our study focused predominantly on a single modality of data, namely the ECG. Although this was partially motivated by the presence of publicly available data, incorporating additional modalities that are routinely collected in a clinical setting would increase the value of our deep-learning system. This could be investigated in future work. Third, our deep-learning system operated at the level of single-lead ECGs and returned a single cardiac arrhythmia label for such ECGs. However, in-hospital settings commonly deal with 12-lead devices. Moreover, multiple cardiac arrhythmias can be present within the same ECG recording. Incorporating this multi-input, multi-output information into our deep-learning system would more reliably reflect clinical settings and medical conditions. This could also be investigated in future work.

Our results demonstrated the utility of a continual deep-learning system focused on cardiac arrhythmia classification based on single-lead ECGs. Prospective validation of such a system would still be required before any potential deployment amongst human patients. Looking forward, our approach can be extended to investigate further notions of task similarity[20,21]. The exploration of more robust definitions of task similarity, their validation through medical domain knowledge and exploitation for generalization are extensions that can add significant value to the intepretability of decisions made by deep-learning systems. An additional extension revolves around predicting destructive interference. At present, destructive interference is often dealt with in a reactive manner. By predicting the degree of forgetting that a system may experience once trained sequentially, we can begin to more proactively alleviate this phenomenon.

## Methods
The two ideas underlying our proposal are the storage of instances into, and the acquisition of instances from, a buffer such that destructive interference is mitigated. We describe these in more detail below.

**Importance-guided buffer storage**. We aim to populate a buffer, $\mathcal{D}_B$, of finite size, $\mathcal{M}$, with instances from the current task that are considered important. To quantify importance, we learn parameters, entitled task-instance parameters, $\beta_{ik}$, associated with each instance, $x_{ik}$, in each task, $k$. These parameters play a dual role.

*Loss-weighting mechanism*. For the current task, $k$, and its associated data, $\mathcal{D}_k$, we incorporate $\beta$ as a coefficient of the loss, $\mathcal{L}_{ik}$, incurred for each instance, $x_{ik} \in \mathcal{D}_k$. For a mini-batch of size, $B$, that consists of $B_k$ instances from the current task, the objective function is shown in Eq. (2). We can learn the values of $\beta_{ik}$ via gradient descent, with some learning rate, $\eta$, as shown in Eq. (3).

$$\mathcal{L} = \frac{1}{B_k} \sum_{i=1}^{B_k} \beta_{ik} \mathcal{L}_{ik} \tag{2}$$

$$\beta_{ik} \leftarrow \beta_{ik} - \eta \frac{\partial \mathcal{L}}{\partial \beta_{ik}} \tag{3}$$

Note that $\frac{\partial \mathcal{L}}{\partial \beta_{ik}} = \mathcal{L}_{ik} > 0$. This suggests that instances that are hard to classify ($\uparrow \mathcal{L}_{ik}$) will exhibit $\downarrow \beta_{ik}$. From this perspective, $\beta_{ik}$ can be viewed as a proxy for instance difficulty. However, as presented, $\beta_{ik} \rightarrow 0$ as training progresses, an observation we confirmed empirically. Since $\beta_{ik}$ is the coefficient of the loss, $\mathcal{L}_{ik}$, this implies that the network will quickly be unable to learn from the data. To avoid this behavior, we initialize $\beta_{ik} = 1$ in order to emulate a standard loss function and introduce a regularization term to penalize its undesirable and rapid decay toward zero. As a result, our modified objective function is:

$$\mathcal{L}_{\text{current}} = \frac{1}{B_k} \sum_{i=1}^{B_k} \beta_{ik} \mathcal{L}_{ik} + \lambda(\beta_{ik} - 1)^2 \tag{4}$$

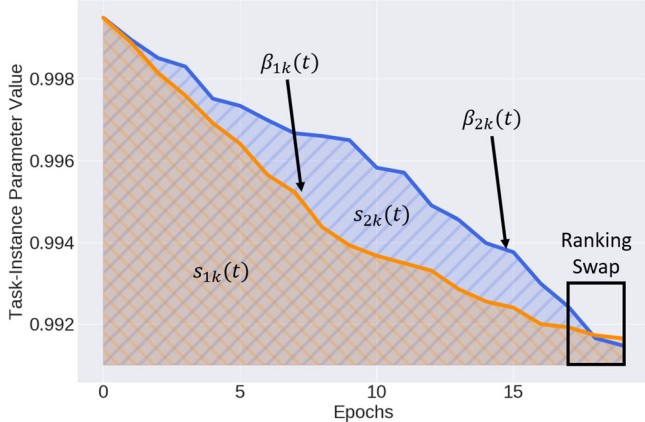

**Fig. 6 Trajectory of $\beta_{1k}$ and $\beta_{2k}$ on task $k$.** Ranking instances based on $\beta$ ($t = 20$) leads to erroneous estimates of their relative difficulty. We propose to rank instances based on the area under the trajectory of $\beta$, denoted as $s_{ik}$.

When $k > 1$, we replay instances from previous tasks by using a replay buffer (see section on buffer acquisition for replay mechanism). These replayed instances incur a loss $\mathcal{L}_{ij} \forall j \in [1 \dots k-1]$. We decide to not weight these instances, in contrast to what we perform to instances from the current task (see Supplementary Table 6).

$$\mathcal{L}_{\text{replay}} = \frac{1}{B - B_k} \sum_{j=1}^{k-1} \sum_{i}^{B_j} \mathcal{L}_{ij} \tag{5}$$

$$\mathcal{L} = \mathcal{L}_{\text{current}} + \mathcal{L}_{\text{replay}} \tag{6}$$

*Buffer-storage mechanism*. We leverage $\beta$, as a proxy for instance difficulty, to store instances into the buffer. To describe the intuition behind this process, we illustrate, in Fig. 6, the trajectory of $\beta_{1k}$ and $\beta_{2k}$ associated with two instances, $x_{1k}$ and $x_{2k}$, while training on the current task, $k$, for $\tau = 20$ epochs. In selecting instances for storage into the buffer, we can (1) retrieve their corresponding $\beta$ values at the conclusion of the task, i.e., at $\beta(t = 20)$, (2) rank all instances based on these $\beta$ values, and (3) acquire the top $b$ fraction of instances. This approach, however, can lead to erroneous estimates of the relative difficulty of instances, as explained next.

In Fig. 6, we see that $\beta_{2k} > \beta_{1k}$ for the majority of the training process, indicating that $x_{2k}$ had been easier to classify than $x_{1k}$. The swap in the ranking of these $\beta$ values that occurs toward the end of training in addition to myopically looking at $\beta$ ($t = 20$) would erroneously make us believe that the opposite was true. Such convergence or swapping of $\beta$ values has also been observed by[22]. As a result, the reliability of $\beta$ as a proxy of instance difficulty is eroded.

To maintain the reliability of this proxy, we propose to track the $\beta$ values after each training epoch, $t$, until the final epoch, $\tau$, for the task at hand and calculate the area under these tracked values. We do so by using the trapezoidal rule as shown in Eq. (7). We explored several variants of the storage function and found the proposed form to work best (see Supplementary Fig. 7). At $t = \tau$, we rank the instances in descending order of $s_{ik}$ (easy to hard) as we found this preferable to the opposite order (see Supplementary Fig. 8), select the top $b$ fraction, and store them into the buffer, in which each task is allotted a fixed portion. The higher the value of the storage fraction, $b$, the more likely it is that the buffer will contain representative instances and thus mitigate forgetting; however, this comes at an increased computational cost.

$$s_{ik} = \int_0^\tau \beta_{ik}(t) dt \approx \sum_{t=0}^\tau \left( \frac{\beta_{ik}(t + \Delta t) + \beta_{ik}(t)}{2} \right) \Delta t \tag{7}$$

**Uncertainty-based buffer acquisition**. During the learning process, a learning system is more likely to benefit from instances that are closer to the decision boundary than those that are farther away[23], an observation primarily established in the active-learning literature. By definition, instances that are close to the decision boundary are those that confuse the system, and which the system might be uncertain about how to classify them. Therefore, to identify instances close to the decision boundary, one can leverage uncertainty-based acquisition functions, such as Bayesian Active Learning by Disagreement (BALD) alongside Monte Carlo Dropout (MCD)[24,25], that quantify a system's uncertainty about a particular instance. We adapt BALD, and experiment with other acquisition functions (see Supplementary Table 8), for use as a buffer-acquisition mechanism in the context of CL.

To quantify the uncertainty of an instance with MCD, the following steps are taken. First, the system receives an instance as an input and, in that process, applies a stochastic binary mask to one or more of its intermediate representations. This generates a posterior probability distribution over a set of classes (cardiac

arrhythmias, in our case). Such a process is repeated $T$ times, applying a different stochastic binary mask each time. These stochastic binary masks can be thought of as perturbations of the parameter space of the network. An instance for which parameter perturbations result in drastic changes to the corresponding posterior probability distribution is likely to be in proximity to the decision boundary. Quantifying these changes is exactly what BALD attempts to do. We explain this more formally next.

At epoch number, $\tau_{MC}$, which we refer to as Monte Carlo (MC) epochs, each of the $M$ instances, $\boldsymbol{x} \sim \mathcal{D}_B$, in the buffer is passed through the network and exposed to a stochastic binary dropout mask to generate a posterior probability distribution, $p(y|\boldsymbol{x}, \boldsymbol{\omega}) \in \mathbb{R}^C$. This is repeated $T$ times to form a matrix, $G \in \mathbb{R}^{M \times T \times C}$. An acquisition function, such as BALD$_{MCD}$, is thus a function $\mathcal{F} : \mathbb{R}^{M \times T \times C} \to \mathbb{R}^M$. It involves calculating the Jensen–Shannon Divergence of the $T$ probability distributions, $\{p_i\}_{i=1}^T$, as shown below.

$$\begin{aligned} \text{BALD}_{MCD} &= \text{JSD}(p_1, p_2, \dots, p_T) \\ &= \text{H}\big(p(y|\boldsymbol{x})\big) - \mathbb{E}_{p(\boldsymbol{\omega}|D_{\text{train}})}\big[\text{H}\big(p(y|\boldsymbol{x}, \hat{\boldsymbol{\omega}})\big)\big] \end{aligned} \quad (8)$$

where $\text{H}(p(y|\boldsymbol{x}))$ represents the entropy of the network posterior probability distributions averaged across the MC samples, and $\hat{\boldsymbol{\omega}} \sim p(\boldsymbol{\omega}|D_{\text{train}})$ is defined as in Gal et al.[25]. At sample epochs, $\tau_S$, we rank instances in descending order of BALD$_{MCD}$ and acquire the top $a$ fraction from each task in the buffer. A higher value of this acquisition fraction, $a$, implies more instances are acquired. Although this may not guarantee improvement in performance, it does guarantee increased training overhead. Nonetheless, the intuition is that by acquiring instances, from previous tasks, to which a network is most confused, it can be nudged to avoid destructive interference in a data-efficient manner. We outline the entire training procedure in Algorithms 1–4 in Supplementary Note 1.

**Baseline methods**. We compare our proposed method to the following. MTL[26] is a strategy whereby all datasets are assumed to be available at the same time and thus can be simultaneously used for training. Although this assumption may not hold in clinical settings due to the nature of data collection, privacy or memory constraints, it is nonetheless a strong baseline. Fine-tuning is a strategy that involves updating all parameters when training on subsequent tasks as they arrive without explicitly dealing with catastrophic forgetting. We also adapt two replay-based methods for our scenarios. GEM[1] solves a quadratic programming problem to generate parameter gradients that do not increase the loss incurred by replayed instances. MIR[2] replays instances from a buffer that incur the greatest change in loss given a parameter pseudo-update. Details on how these methods were adapted are found in Supplementary Note 3.

**Evaluation metrics**. To evaluate our methods, we exploit metrics suggested by[1] such as average AUC and BWT. We also propose two additional evaluation metrics that provide us with a more fine-grained analysis of learning strategies.

*t-Step backward transfer*. To determine how performance changes "t-steps into the future," we propose BWT$_t$ that evaluates the performance of the network on a previously seen task (for a total of $N$ tasks), after having trained on t subsequent tasks.

$$\text{BWT}_t = \frac{1}{N-t} \sum_{j=1}^{N-t} \text{R}_j^{j+t} - \text{R}_j^j \quad (9)$$

*Lambda backward transfer*. We extend BWT$_t$ to all time-steps, $t$, to generate BWT$_\lambda$. As a result, we can identify improvements in methodology at the task level.

$$\text{BWT}_\lambda = \frac{1}{N-1} \sum_{j=1}^{N-1} \left[ \frac{1}{N-j} \sum_{t=1}^{N-j} \text{R}_j^{j+t} - \text{R}_j^j \right] \quad (10)$$

**Hyperparameters**. Depending on the CL scenario, we chose $\tau = 20$ or $40$, as we found that to achieve strong performance on the respective validation sets. We chose $\tau_{MC} = 40 + n$ and the sample epochs $\tau_S = 41 + n$ where $n \in \mathbb{N}^+$ in order to sample data from the buffer at every epoch following the first task. The values must satisfy $\tau_S \geq \tau_{MC} > \tau$. For computational reasons, we chose the storage fraction $b = 0.25$ of the size of the training dataset and the acquisition fraction $a = 0.50$ of the number of samples per task in the buffer. To calculate the acquisition function, we chose the number of MC samples, $T = 20$. We chose the regularization coefficient, $\lambda = 10$. We also explore the effect of changing these values on performance (see Supplementary Note 4).

**Reporting summary**. Further information on research design is available in the Nature Research Reporting Summary linked to this article.

## Data availability

The data used in this study are freely and publicly available. The Cardiology dataset can be accessed at https://irhythm.github.io/cardiol_test_set/. The Chapman dataset can be accessed at https://figshare.com/collections/ChapmanECG/4560497/2. The PhysioNet 2020 dataset can be accessed at https://physionetchallenges.org/2020/. The PhysioNet 2017 dataset can be accessed at https://physionet.org/content/challenge-2017/1.0.0/.

## Code availability

All models were developed using Python and standard deep-learning frameworks such as PyTorch[27]. Code for pre-processing the data can be found at https://github.com/danikiyasseh/loading-physiological-data. The code for running the experiments can be found at https://github.com/danikiyasseh/CLOPS.

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

## Acknowledgements
We thank Abdel Halim Hafez and Farid Al-Atrash for lending us their voice. D. C. was supported by the EPSRC under Grants EP/P009824/1 and EP/N020774/1, and by the National Institute for Health Research (NIHR) Oxford Biomedical Research Centre (BRC). The views expressed are those of the authors and not necessarily those of the NHS, the NIHR, or the Department of Health. T.Z. was supported by the Engineering for Development Research Fellowship provided by the Royal Academy of Engineering.

## Author contributions
D.K. conceived and designed the study, performed the data analysis, and prepared the manuscript. D.K. and T.Z. checked the validity of the data. All authors contributed to results interpretation and final manuscript preparation.

## Competing interests
The authors declare no competing interests.
