## [Peer Review File · Nature Communications]

Reviewers' Comments:

Reviewer #1:

Remarks to the Author:

In this work, Kiyasseh et al. propose a Continual learning strategy to address the problem of model update under data distribution shift or across various related but distinct tasks. The key issue is to maintain acceptable model performance in a dynamically changing environment without 'catastrophic forgetting' the older tasks. This work focused on the task of 'cardiac arrhythmia diagnosis' in several dynamic environments. Learning without forgetting is achieved via replaying important tasks (data points/labels) from the past, prioritized by task difficulty and uncertainty. This is a pioneering work on the application of continuous learning to clinical data. I have a number of questions/comments.

1) The experimental setup is interesting (Class-IL, Time-IL, Domain-IL), however, the single most significant challenge in this domain is building generalizable models across different institutions. Continual learning across different datasets (institutions) is particularly challenging due to HIPAA/GDPR which makes the idea of moving data across institutional boundaries for 'replay' subject to privacy constraints. It would have been good to see examples of the performance of CLOPS across institutions with potentially 'differential privacy' provisions put in place.

2) Under the Class-IL problem setting, it is not clear if a single multimodal classifier or multiple binary classifiers were used. Are you swapping in-and-out multiple binary classifiers at the last layer depending on the task?

3) How do you address the issue of label noise (or potentially adversarial examples) across various tasks? Is there a way to detect new instances that are totally incompatible with what the AI system learned in the past?

4) In Equation 1, it is not clear why the loss for a given instance could not be used directly as a measure of task difficulty. It is also not clear that a 'difficult' task should be replayed again and again through the network, given the potential for label noise (incorrect labels) from the prior tasks.

5) It seems to me that Beta is not just a function of task difficulty, it also a function of label prevalence/incidence rates across various tasks. Do you use balanced labels within your mini-batches?

6) Utilization of uncertainty (instances that fall close to decision boundary) is interesting. But, how confident are you on your method of uncertainty estimation?* More information on the MC approach will be useful (potentially in an appendix).

* Variational inference is known to underestimate uncertainty

Blei, D. M., Kucukelbir, A. & McAuliffe, J. D. Variational inference: a review for statisticians. *J. Am. Stat. Assoc.* 112, 859–877 (2017)

Reviewer #2:

Remarks to the Author:

Summary:

The authors present an application of rehearsal-based Continual Learning methods to the diagnosis of cases from cardiac signals. To the best of my knowledge, the authors apply such techniques to the problem domain for the first time, possibly making valuable contributions to a domain of great importance to society. From a methodological standpoint the proposed ideas are simple and would

not necessarily be considered to be sufficiently novel to be accepted for publication as a general purpose solution to the Continual Learning problem. Hence, this work should in my opinion be regarded as an application paper which may well have impact to the concrete problem at hand but will have limited impact to continual learning problems in other domains.

Detailed comments:

Abstract:

- "In three continual learning scenarios based on three publically-available datasets, we show that CLOPS can outperform the state-of-the-art methods" - This is slightly misleading as it can be understood as the results in Table 1 suggest that CLOPS outperforms the considered methods in merely two of the three scenarios. Specifically, MIR outperforms CLOPS in Time Incremental Learning.

Introduction:

- Line 25-26: I wouldn't necessarily agree that Continual Learning is a problem specific to Deep Learning, nor that the solution must be a Deep Learning method. I suggest simply writing "learning system"
- Line 37-41: I'm not necessarily convinced there is sufficient evidence that this is indeed the case. I suggest explicitly stating that this is a hypothesis by the authors, not a generally accepted fact.
- Line 42-47: The review of past work in Continual Learning misses much important and relevant research that I would like to see cited, as this is likely to be read by researchers who might not be deeply familiar with Continual Learning.
- Line 67-68: It seems unlikely to me that the idea of learning a weighting of each datapoint in the loss function does not already exist in the literature. An example are the role dual variables play in Support vector machines. I suggest conducting a thorough literature review to find connected ideas. This might even provide inspiration to further improvements to the system.

Evaluation:

- Line 185-186 and elsewhere (minor): BWT stands for backward transfer, not backward weight transfer
- Table 1:
 - Class Incremental Learning: Why does Fine-tuning outperform GEM, MIR and MTL? This seems highly unlikely to me and is in contradiction to much research in CL that consistently shows GEM & MIR outperforming Fine-tuning.
 - Time Incremental Learning: What happened with GEM? Also, the authors do not explicitly refer to the negative result (MIR outperforming CLOPS) in the text.
 - Figure 2 & 3: These are rather difficult to read and could be presented in a neater fashion. An example would be a separate subfigure for each task to avoid too many overlapping colours. At the current stage, the figures look hastily put together with little care for appropriate graphic design.
- Line 266-268: I'm not convinced that this type of forward transfer is unique to CLOPS and might be more a result of the data distribution than an inherent feature of the method. This has certainly proven to be the case for existing methods applied to various CL problems.

Validation of Interpretation of Storage Parameter:

- I'm overall not convinced that this set of parameters is necessarily a better proxy than much simpler ideas such as considering the per-example loss as a measure of "difficulty". Hence, the manuscript lacks a comparison to such simple baselines. The fact that additional regularisation is required to lead to satisfactory results is a strong argument against the method. I suspect that additional cross-validation required to choose λ (Equation 3) also negates the claimed 20-fold training overhead (Line 268)
- Figure 5 is overall quite unsurprising and not particularly interesting. It also fails to explain why

the worst performing task in Figure 2b) (Task 4-5) is rated of medium difficulty when looking at the distribution of importance parameters.

- Line 373-375: As discussed before I'm not convinced that this is particularly useful. Why not simply look at the loss? Importantly, those values are not available without training, and can hence not be computed on a held-out validation or test set

Discussion:

- Line 387-392: This is my biggest concern. I suggest clarifying with domain-experts whether this is a realistic assumption in the application at hand. I appreciate the honesty of the authors to point this out.

Methods:

- The problems with a native use of the loss-weighting mechanism (Figure 6 and the needed regularisation) are key reasons why I don't think this idea will have large impact beyond the application area. Once again, I suggest the authors investigate alternatives.

Uncertainty-based Buffer Acquisition:

- Please reference related work that has similar Uncertainty-based Buffer Acquisition. Examples are [1], [2].

[1] Titsias, M. K., Schwarz, J., Matthews, A. G. D. G., Pascanu, R., & Teh, Y. W. (2019). Functional regularisation for continual learning with gaussian processes. arXiv preprint arXiv:1901.11356.

[2] Nguyen, C. V., Li, Y., Bui, T. D., & Turner, R. E. (2017). Variational continual learning. arXiv preprint arXiv:1710.10628.

We would like to thank the reviewers for taking the time and effort to review our manuscript and for providing us with valuable feedback. We have addressed your comments below and modified the manuscript accordingly.

Reviewer 1

In this work, Kiyasseh et al. propose a Continual learning strategy to address the problem of model update under data distribution shift or across various related but distinct tasks. The key issue is to maintain acceptable model performance in a dynamically changing environment without 'catastrophic forgetting' the older tasks. This work focused on the task of 'cardiac arrhythmia diagnosis' in several dynamic environments. Learning without forgetting is achieved via replaying important tasks (data points/labels) from the past, prioritized by task difficulty and uncertainty. This is a pioneering work on the application of continuous learning to clinical data. I have a number of questions/comments.

Comment 1

The experimental setup is interesting (Class-IL, Time-IL, Domain-IL), however, the single most significant challenge in this domain is building generalizable models across different institutions. Continual learning across different datasets (institutions) is particularly challenging due to HIPAA/GDPR which makes the idea of moving data across institutional boundaries for 'replay' subject to privacy constraints. It would have been good to see examples of the performance of CLOPS across institutions with potentially 'differential privacy' provisions put in place.

We agree with the reviewer's comments regarding the importance of models that generalize *across* institutions. To that end, we have introduced a new and fourth continual learning scenario, entitled **Institute Incremental Learning**. In this scenario, we task the system with performing cardiac arrhythmia diagnosis in response to changes in healthcare institutions. These changes manifested in the form of tasks comprising completely different datasets. Specifically, the system was trained on the following datasets in sequence: Chapman, PhysioNet 2017, and Cardiology. Please refer to the *continual learning scenarios* section of the modified manuscript for more details.

Although preserving patient privacy and addressing operational challenges are of importance within the clinical machine learning community, we believe these topics are beyond the scope of our research. Nonetheless, we have emphasized their importance in the *discussion* section of the modified manuscript.

Comment 2

Under the Class-IL problem setting, it is not clear if a single multimodal classifier or multiple binary classifiers were used. Are you swapping in-and-out multiple binary classifiers at the last layer depending on the task?

In the Class-IL scenario, we are using multiple binary classifiers, as you suggested. For clarity, we have included a statement about this in the *continual learning scenarios* section of the modified manuscript. Please note that although the deep-learning system is task-specific in the Class-IL scenario, it is *task-agnostic* in the remaining continual learning scenarios. This implies that the system is not aware of the task identity of the data it receives. We have also modified Fig. 1 to reflect this.

Comment 3

How do you address the issue of label noise (or potentially adversarial examples) across various tasks? Is there a way to detect new instances that are totally incompatible with what the AI system learned in the past?

As for label noise, we believe that the degree to which it will impact the continual learning system depends on a multitude of factors including the continual learning scenario (Class-IL vs. Time-IL vs. Domain-IL), the absolute amount of data stored in the buffer, and the amount of data retrieved from the buffer. For example, storing in the buffer a small subset of instances suffering from label noise is likely to hinder the replay process in future tasks. However, in light of our importance-based storage mechanism, we hypothesize that such instances are unlikely to be stored in the first place. To understand this better, let us consider the following example. Let us assume we have two instances, x_1 and x_2 , where x_1 has an incorrect ground-truth annotation. As a result of this label noise, the deep-learning system is likely to have a harder time classifying x_1 than x_2 . Based on our setup, this implies that the task-instance parameter, β_{x_1} will decay much faster than β_{x_2} , and thus have a lower area under the trajectory, $s_{x_1} < s_{x_2}$. Since we acquire instances with the *highest* s values, we are unlikely to select noisy instances for storage. This intuition is further supported by findings in <https://openreview.net/pdf?id=HyenUkrtDB>.

Comment 4

In Equation 1, it is not clear why the loss for a given instance could not be used directly as a measure of task difficulty. It is also not clear that a 'difficult' task should be replayed again and again through the network, given

the potential for label noise (incorrect labels) from the prior tasks.

As a point of clarification, when deciding which instances to store into the buffer, we choose the *least* difficult instances. In fact, we empirically showed (in Supplementary Fig. 8 on page 13) that this approach is preferable, from a generalization perspective, to storing the *most* difficult instances. This could be because these instances have incorrect ground-truth annotations, as the reviewer suggested, or that they were simply too difficult to be diagnosed by the deep-learning system.

To more concretely address the reviewer's comments regarding the apparent inter-changeability of our task-instance parameters (β) and the per-instance loss, we have dedicated an entire section in the modified manuscript to exploring their differences. Please refer to the section entitled *Exploration of alternative storage mechanisms*. In short, we provide both qualitative and quantitative evidence to elucidate the differences between task-instance parameters and the per-instance loss.

Comment 5

It seems to me that Beta is not just a function of task difficulty, it also a function of label prevalence/incidence rates across various tasks. Do you used balanced labels within your mini-batches?

Based on Eq. 1, we see that β is a function of the loss. The loss, in turn, can be a function of a multitude of factors including but not limited to task difficulty, (potential) label noise, the order in which data are presented to a deep-learning system, and the ratio of data from each class/label. More broadly, with clinical data, labels are unlikely to be balanced given the higher incidence of *normal* cases relative to *abnormal* ones. In light of all these factors, we do not balance labels within our mini-batches.

Comment 6

Utilization of uncertainty (instances that fall close to decision boundary) is interesting. But, how confident are you on your method of uncertainty estimation?* More information on the MC approach will be useful (potentially in an appendix).

Our uncertainty-based buffer acquisition mechanism is based on an approach with an established track record in the active-learning literature. It leverages Monte Carlo Dropout (MCD) alongside an acquisition function known as Bayesian Active Learning by Disagreement (BALD). We have provided an in-depth description of this approach, and its underlying intuition, in the section entitled *Uncertainty-based buffer acquisition*. We also compared this particular acquisition function to other variants (see Supplementary Table 8 on page 20) and found it to be favourable.

Reviewer 2

The authors present an application of rehearsal-based Continual Learning methods to the diagnosis of cases from cardiac signals. To the best of my knowledge, the authors apply such techniques to the problem domain for the first time, possibly making valuable contributions to a domain of great importance to society. From a methodological standpoint the proposed ideas are simple and would not necessarily be considered to be sufficiently novel to be accepted for publication as a general purpose solution to the Continual Learning problem. Hence, this work should in my opinion be regarded as an application paper which may well have impact to the concrete problem at hand but will have limited impact to continual learning problems in other domains.

Comment 1

"In three continual learning scenarios based on three publically-available datasets, we show that CLOPS can outperform the state-of-the-art methods" - This is slightly misleading as it can be understood as the results in Table 1 suggest that CLOPS outperforms the considered methods in merely two of the three scenarios. Specifically, MIR outperforms CLOPS in Time Incremental Learning.

We have modified the abstract and other relevant sections of the manuscript to avoid potentially misleading statements and to be more precise in our claims.

Comment 2

- Line 25-26: I wouldn't necessarily agree that Continual Learning is a problem specific to Deep Learning, nor that the solution must be a Deep Learning method. I suggest simply writing "learning system"

We have modified this sentence accordingly.

Comment 3

Line 37-41: I'm not necessarily convinced there is sufficient evidence that this is indeed the case. I suggest explicitly stating that this is a hypothesis by the authors, not a generally accepted fact.

We have modified this sentence accordingly.

Comment 4

Line 42-47: The review of past work in Continual Learning misses much important and relevant research that I would like to see cited, as this is likely to be read by researchers who might not be deeply familiar with Continual Learning.

We have expanded our discussion of related work in the domain of continual learning. We have also referenced a great review of such methodologies that was recently published by Parisi *et al.*. Please see the introduction in the modified manuscript.

Comment 5

Line 67-68: It seems unlikely to me that the idea of learning a weighting of each datapoint in the loss function does not already exist in the literature. An example are the role dual variables play in Support vector machines. I suggest conducting a thorough literature review to find connected ideas. This might even provide inspiration to further improvements to the system.

We agree that learning an instance weighting is a relatively established technique in the machine learning community. However, we believe that our contribution stems from *how* we learn these weightings (via stochastic gradient descent), and in what capacity they are used (to guide the storage of instances into a buffer in the context of continual learning). If we were to contrast this approach with that of SVMs, we would see that in the latter, the dual variables are learned by solving a quadratic program and are used to determine which instances are support vectors.

Comment 6

Line 185-186 and elsewhere (minor): BWT stands for backward transfer, not backward weight transfer

We have modified this sentence and others accordingly.

Comment 7

- Class Incremental Learning: Why does Fine-tuning outperform GEM, MIR and MTL? This seems highly unlikely to me and is in contradiction to much research in CL that consistently shows GEM MIR outperforming Fine-tuning

Recall that the Class-IL scenario involves a sequence of binary classification tasks. However, the multi-task learning (MTL) method involves training on all data, from all classes, at the same time. This becomes a multi-class classification setting, which is relatively more difficult to solve than just a binary classification task. This explains the lower MTL score relative to Fine-tuning. We have mentioned this in the *evaluation* section of the modified manuscript. As for the relatively poorer performance of GEM and MIR, we hypothesize that this could be due to two factors. The first is that these methods were not originally designed for, nor validated on, clinical time-series data. Instead, they were designed for the domain of computer vision. As a result, findings in such domains may not generalize cleanly to clinical data. Second, recall that the results in Table 1 are for a particular storage and acquisition fraction. In Supplementary Table 9 (on page 21), we found that this acquisition fraction has a significant impact on the performance of MIR.

Comment 8

Time Incremental Learning: What happened with GEM? Also, the authors do not explicitly refer to the negative result (MIR outperforming CLOPS) in the text.

The GEM algorithm involves solving a quadratic programming problem, in the event a certain inequality is violated. Upon implementing the GEM algorithm in our Time-IL and Institute-IL scenarios, the aforementioned quadratic programming problem is unable to be solved; namely, it returns an error which we did not experience when conducting the Class-IL and Domain-IL experiments. As for explicitly discussing negative results, we have taken a more pro-active approach to doing so in the modified manuscript.

Comment 9

Figure 2 3: These are rather difficult to read and could be presented in a neater fashion. An example would be a separate subfigure for each task to avoid too many overlapping colours. At the current stage, the figures look hastily put together with little care for appropriate graphic design.

As a point of reference, our design of these figures was inspired by the figures in the following continual learning paper by researchers at DeepMind <https://arxiv.org/pdf/1811.11682.pdf>. The purpose of the plot is to illustrate the degree of catastrophic forgetting that takes places when transitioning from one task to the next. Nonetheless, in the modified manuscript, to improve the clarity of these figures, we have enlarged them and updated the corresponding caption.

Comment 10

Line 266-268: I'm not convinced that this type of forward transfer is unique to CLOPS and might be more a result of the data distribution than an inherent feature of the method. This has certainly proven to be the case for existing methods applied to various CL problems.

We agree that forward transfer may occur 'naturally' when a network is exposed to a sequence of tasks if, for example, the adjacent tasks are similar to one another, or if the network is transitioning from an easy to hard task (as in curriculum learning). However, if this natural forward transfer were *independent* of our method, and dependent on the data distribution, we would expect to at least see signs of it in the Fine-tuning experiment. However, that is not the case. We have included this discussion in the *evaluation* section.

Comment 11

I'm overall not convinced that this set of parameters is necessarily a better proxy than much simpler ideas such as considering the per-example loss as a measure of "difficulty". Hence, the manuscript lacks a comparison to such simple baselines. The fact that additional regularisation is required to lead to satisfactory results is a strong argument against the method. I suspect that additional cross-validation required to choose λ (Equation 3) also negates the claimed 20-fold training overhead (Line 268)

To address the reviewer's comments regarding the apparent inter-changeability of our task-instance parameters (β) and the per-instance loss, we have dedicated an entire section to exploring their differences. This section is entitled *Exploration of alternative storage mechanism*. In short, we experiment with a different storage mechanism where, instead of tracking our task-instance parameters, we track the per-instance loss and exploit that to identify instances for storage into the buffer. We ultimately show that despite the simple interpretation of task-instance parameters as a proxy for instance difficulty, they confer higher benefits to generalization performance and backward transfer relative to a storage strategy based on the per-instance loss.

As for the act of regularization more broadly, it has become an aspect of the training of many deep-learning systems that we now take for granted despite its importance, e.g., with traditional L_1 and L_2 regularization to mitigate the effects of over-fitting. In light of this, we believe that regularization is a simple tool that we can leverage during training.

As for our specific claim regarding the potential reduced overhead, we understand that additional factors play a role in contributing to that. As a result, we have modified the sentence claiming a 20-fold reduction in training overhead such that it poses that as a possibility and not a guarantee.

Comment 12

Figure 5 is overall quite unsurprising and not particularly interesting. It also fails to explain why the worst performing task in Figure 2b) (Task 4-5) is rated of medium difficulty when looking at the distribution of importance parameters.

It might be tempting to correlate the relative difficulty of tasks (as captured by Fig. 4) to their relative performance (as shown in Fig. 2). We believe that such attempts might be of little value given the absence of a strict correlation between loss values and performance scores, such as the AUC. In other words, a decrease in the loss does not always translate to a higher AUC score.

We believe Fig. 5 adds value in two ways. First, as we have shown in the main manuscript, it allows us to qualitatively validate our interpretation of task-instance parameters at the *instance* level. Instances with a low s value are rightly confusing to the network whereas those with a high s value are relatively straightforward to diagnose. Second, in light of the appearance of the various distributions in Fig. 4, we were motivated to quantitatively validate our interpretation of the task-instance parameters. This approach involves exploiting the distributions to quantify task similarity and set up a learning curriculum. Please see the section entitled *Validation of interpretation of storage parameters* in the modified manuscript for an in-depth description of this approach.

Comment 13

Line 373-375: As discussed before I'm not convinced that this is particularly useful. Why not simply look at the loss? Importantly, those values are not available without training, and can hence not be computed on a held-out validation or test set

By looking solely at the loss incurred when classifying an instance, practitioners are unable to observe the learning dynamics (over time) of the network. In contrast, by tracking our task-instance parameters, this dynamic capability is allowed. Knowledge of such dynamics can allow researchers, for example, to identify whether certain instances were diagnosed more easily at the beginning of training and less easily at the end.

One could also argue that the loss can be tracked, analogously to our task-instance parameters. Although that may be a valid approach, task-instance parameters present a key difference to loss terms. In addition to their buffer-storage capabilities, they act as a weighting mechanism on the per-instance loss terms. This means that, during training, the amount the model learns from each instance differs (e.g., it learns more if β is higher). In addition to clarifying this point in the modified manuscript, we refer the reviewer to our response to Comment 11 and the new section entitled *Exploration of alternative storage mechanism* in the manuscript.

Comment 14

Line 387-392: This is my biggest concern. I suggest clarifying with domain-experts whether this is a realistic assumption in the application at hand. I appreciate the honesty of the authors to point this out.

After discussing this issue with a domain expert, we both agreed that patient privacy is of utmost importance in clinical settings, and that as long as such a model was deployed within the confines of a health system, it should not pose an obstacle. Nonetheless, as we outlined to Reviewer 1, we believe there exist promising paths for the future such as designing systems that incorporate differential privacy. These explorations are beyond the scope of our work.

Comment 15

The problems with a native use of the loss-weighting mechanism (Figure 6 and the needed regularisation) are key reasons why I don't think this idea will have large impact beyond the application area. Once again, I suggest the authors investigate alternatives.

Our manuscript is indeed focused on continual learning for cardiac time-series signals. We have shown that our storage mechanism (based on task-instance parameters) adds value, from a model generalization perspective, relative to a random storage mechanism (Fig. 3). Nonetheless, to allay further concerns, we take the reviewers' advice and investigate an alternative storage mechanism involving the per-instance loss (see the section entitled *Exploration of alternative storage mechanism*). We still find that our proposed task-instance parameters are preferable to this alternative approach. Moreover, in the manuscript, we do not make generalizations to other application areas. In fact, we suggest incorporating multiple data modalities as a future path worth exploring. If we were to follow this path, one would think that a modification as simple as a loss-weighting mechanism could be easily adopted by other researchers, and thus might have an impact on other application areas.

Comment 16

Please reference related work that has similar Uncertainty-based Buffer Acquisition. Examples are [1], [2].

Thank you for directing us to these references. We have now included them in the introduction when discussing related work.

Reviewers' Comments:

Reviewer #1:

Remarks to the Author:

Thank you for your thorough response. I have no further comments.

Reviewer #2:

Remarks to the Author:

I've read the authors revised manuscript and would like to thank them for the effort taken to address the reviewers comments. I was pleased to see most of my points addressed and in particular appreciated the "Exploration of alternative storage mechanism" section being added.

We would like to thank the reviewers for taking the time and effort to review our manuscript and for providing us with valuable feedback.

Reviewer 1

Thank you for your thorough response. I have no further comments.

Reviewer 2

I've read the authors revised manuscript and would like to thank them for the effort taken to address the reviewers comments. I was pleased to see most of my points addressed and in particular appreciated the "Exploration of alternative storage mechanism" section being added.